# Analysis of HPV Vaccination Willingness amongst HIV-Negative Men Who Have Sex with Men in China

**DOI:** 10.3390/vaccines9101069

**Published:** 2021-09-24

**Authors:** Wei He, Haiying Pan, Bing Lin, Xiaoni Zhong

**Affiliations:** School of Public Health and Management, Chongqing Medical University, Chongqing 400016, China; 2020111423@stu.cqmu.edu.cn (W.H.); 2020111428@stu.cqmu.edu.cn (H.P.); 2019110918@stu.cqmu.edu.cn (B.L.)

**Keywords:** MSM, HPV, vaccine, vaccination willingness

## Abstract

Objective: Men who have sex with men (MSM) are high-risk groups of human papillomavirus (HPV) infection, the best measure to prevent this is the HPV vaccine. However, few studies have investigated HPV vaccination willingness in the MSM population in China. We aimed to explore the willingness of human immunodeficiency virus (HIV)-negative MSM for HPV vaccination and the factors affecting their willingness to vaccinate. Methods: We adopted a non-probability sampling method to recruit HIV-negative MSM volunteers. Participants completed a questionnaire, including sociodemographic characteristics, sexual behavior characteristics, HPV infection and vaccine-related knowledge, risk perception, and HPV vaccination willingness and promotion attitudes. Results: Of the 406 HIV-negative MSM surveyed, 86.21% were willing to receive HPV vaccine. HPV infection and vaccine-related knowledge (odds ratio [OR] = 2.167, 95% confidence interval [CI] = 1.049–4.474), HPV infection risk perception (OR = 5.905, 95% CI = 1.312–26.580), and HPV vaccine promotion attitude (OR = 6.784, 95% CI = 3.164–14.546) were all related to HPV vaccination willingness. Conclusion: MSM have a high willingness for HPV vaccination. Strengthening health education for MSM, raising their awareness of HPV infection and vaccines, and promoting their risk perception of HPV infection will help increase their willingness for HPV vaccination.

## 1. Introduction

Human papillomavirus (HPV) infection is the most common sexually transmitted disease (STD) around the world, and more than half of people who are sexually active have been infected with HPV at least once [1,2]. HPV infection can cause various types of genital warts and malignant lesions such as cervical cancer, penile cancer, and anal cancer [3,4]. At present, more than 200 types of HPV have been identified. According to different carcinogenicity, HPV can be divided into high-risk and low-risk types [5,6].

The most effective way to prevent HPV infection is to receive the HPV vaccine. Currently, three main types of HPV vaccines have been approved for marketing in countries including China, namely, 2-valent, 4-valent, and 9-valent HPV vaccines. Amongst them, 2-valent HPV vaccines can prevent HPV16/18 infections, 4-valent HPV vaccines can prevent HPV6/11/16/18 infection, and 9-valent HPV vaccines can prevent HPV6/11/16/18/31/33/45/52/58 infection [7,8]. These three vaccines have shown good safety and effectiveness in various trials and post-marketing surveillance and can prevent approximately 70–90% of HPV-related cancers [3,9,10]. Because cervical cancer and other HPV-related diseases are important public health issues around the world, the WHO recommends that HPV vaccine be included in the national immunization program. As of 15 February 2019, 92 countries have adopted the HPV vaccine as part of their national vaccination plans.

It is worth noting that about 90% of the 270,000 cervical cancer deaths in 2015 occurred in low-income and middle-income countries (LMIC), where the mortality is 18 times higher than that in developed countries. The lifetime risks (up to age 74 years) of developing cervical cancer were 0.9% for women in high-income countries and 1.6% in LMICs, and the risks of death due to cervical cancer were 0.3% for women in high-income countries and 0.9% in LMICs. The health system in LMICs is very poor to provide corresponding precautions, including the HPV vaccine [11].

The HPV vaccine is currently mainly targeted at 9–45-year-old females. In 2007, the Australian Federal Government began to provide HPV vaccines for girls aged 12 to 13 years for free. In 2016, 78.6% of 15-year-old girls had been vaccinated. Studies found that the prevalence of HPV amongst women aged 18–24 years dropped from 22.7% to 1.1% in 2005–2015 [12,13].

At present, research on HPV vaccines is mainly focused on women, whilst research on men is relatively rare. However, men play the role of ‘sexual transmission carrier’ in the process of female HPV infection. Male HPV infection not only harms their own reproductive health, leading to STDs, infertility, or cancer, but also brings hidden dangers to their female partners. In Western Europe and other developed countries, the burden of HPV-related diseases in men is equivalent to that of women [14,15]. A large number of studies have found that HPV vaccines also have a good preventive effect on malignant tumors such as penile cancer and anal cancer. Countries such as the United States and Australia have also begun to expand the HPV vaccination program to young men [16,17,18]. However, given that China has not completed the HPV vaccine trial in men, it is not recommended to vaccinate men for the time being.

In fact, the HPV vaccination program for women has shown a group effect, and that heterosexual men can also benefit from it. However, men who have sex with men (MSM) are unlikely to benefit from the group effect due to the particularity of their sexual partners [19]. Therefore, in addition to women who are closely related to HPV-related diseases, the MSM population bears a heavier disease burden compared with heterosexual men. In addition, because unprotected oral sex and anal sex are very common in MSM, the risk of HPV infection in MSM is much higher. For example, the incidence of anal cancer in MSM is very high, which is 20 times that of heterosexual men [20]. Experimental evidence shows that HPV vaccination in MSM will be a cost-effective intervention that can reduce the occurrence and recurrence of HPV infection-related diseases such as genital warts and anal cancer [15,18,21,22]. In 2011, the Advisory Committee on Immunization Practices recommended HPV vaccination through age 26 years for MSM and for immunocompromized persons (including those with HIV infection) if not vaccinated previously. The United Kingdom began its national HPV vaccination program for MSM in April 2018 [23].

Compared with LMICs, the United States, Italy, and Australia have great advantages in economy, education, and medical care. In LMICs, people’s weak awareness of protection, the poor economic level of society and the limited health system capacity of the national government are all factors hindering the further promotion of HPV vaccine.

Compared with developed countries, China still needs to make great efforts to promote HPV vaccination. HPV vaccines have been licensed in China since 2016, and they are expensive and only available in limited supply. These HPV vaccines are only for women. As of August, 2020, there is no HPV vaccination program in China [24].

Many studies have evaluated individuals’ knowledge, attitudes, and risk perceptions about HPV vaccination and their relationship with HPV vaccination willingness. According to the theory of knowledge, attitude, and practice (KAP), knowledge is the basis of behavior change, and attitudes and beliefs are the driving force for behavior change. Only when people acquire relevant knowledge and actively think about it can they gradually form beliefs, including risk perception of risk factors. Only when knowledge rises to belief can it be possible to adopt a positive attitude to change behavior [25,26]. Therefore, HPV infection and vaccine-related knowledge, attitudes, and risk perception have been widely regarded as determinants of vaccination willingness [27,28,29]. However, the current research on HPV vaccination willingness of MSM in China is still relatively rare. This study aims to explore the HPV vaccination willingness of MSM and related factors that affect the vaccination willingness, and provide a reference for the future development of HPV vaccination strategies for male populations.

## 2. Materials and Methods

### 2.1. Study Design and Implementation

The survey population in this study came from the cohort of “National Science and Technology Major Project 2018ZX10721102-005”. In this study, participants were recruited in Chongqing and Sichuan provinces using a non-probability sampling method (including a non-governmental organization, peer introductions, human immunodeficiency virus [HIV] voluntary counselling and testing outpatient service, QQ, and WeChat). The recruitment conditions for MSM volunteers are as follows: (1) 18 years of age or older; (2) having had anal or oral sex with the same sex; and (3) a negative HIV antibody test result. After obtaining the informed consent of the volunteers, the staff distributed the structured electronic questionnaire to the volunteers and collected them uniformly. The sample size of this study was estimated on the basis of the current situation survey formula $n=\frac{Z_{\alpha /2}^{2}\left(1-P\right)P}{\delta^{2}}$, *α* = 0.05, *Z*_*α*/2_ = 1.96 ≈ 2. The HPV vaccine acceptance rate of the MSM population was about 56% [30], *P* = 0.56, *δ* = 0.1 ∗ *P*. The required sample size was calculated to be 314 people. In the end, this study investigated 442 HIV-negative MSM populations. This study was approved by the Ethics Committee of Chongqing Medical University (2019001).

### 2.2. Survey Instrument

The questionnaire used in this study was adapted from the questionnaire that has been widely recognized and used in previous studies [31,32,33,34]. The questionnaire has five main parts, including (1) sociodemographic characteristics; (2) sexual behavior characteristics and preventive behaviors; (3) HPV infection and vaccine-related knowledge; (4) risk perception; and (5) HPV vaccination willingness and promotion attitude. There are a total of 12 questions about HPV infection and vaccine-related knowledge. Answering more than nine questions correctly indicates a high level of knowledge, and the rest are medium or low levels.

### 2.3. Statistical Analysis

In this study, frequency and percentage were used to describe the data, and the chi-square test or Fisher’s exact test was used for categorical data. Multivariate analysis adopted stepwise logistic regression analysis. All variables with *p* < 0.2 in single-factor analysis were included in the model, and the odds ratio (OR) and 95% confidence interval (CI) were calculated. Statistical significance was considered at *p* < 0.05. The significance level for entry (Slentry, SLE) was 0.05, and the significance level for stay (Slstay, SLS) was 0.10. We performed the statistical analysis using SAS 9.4 software (SAS Institute, Cary, NC, USA).

## 3. Results

In this study, a total of 442 MSM were surveyed, and some questionnaires with serious quality problems, such as missing and incorrect filling, were excluded. Finally, 406 people were included in the study.

### 3.1. Sociodemographic Characteristics

Amongst the MSM survey population, 42.6% were 18–30 years old, 97.8% had Han nationality, 69.5% had a college degree or above, 74.6% were employed, 14.8% were students, and 74.6% were unmarried. About 54.4% had a monthly income of 1000–5000 yuan (Table 1).

### 3.2. Sexual Behaviour Characteristics and Preventive Behaviours

Amongst the MSM survey population, in the last 6 months, 42.9% had only one male sexual partner (including temporary and long-term sexual partner), whilst 27.6% had multiple sexual partners. The MSM population attached great importance to STD and acquired immunodeficiency syndrome (AIDS). About 74.9% and 62.6% had undergone STD testing and counselling, and 86.2% and 75.4% had undergone AIDS testing and counselling, respectively (Table 2).

### 3.3. Risk Perception

Amongst the MSM-surveyed population, only 19.7% thought that they were more likely to be infected with HPV, 47.0% thought that they were less likely to be infected with HPV, and 42.6% thought that HPV was a greater threat to them. Amongst the MSM who had not been diagnosed with STDs, 57.8% thought that they were less likely to be infected with STDs, whilst only 13.4% thought that they were more likely to be infected with STDs (Table 3).

### 3.4. HPV Infection and Vaccine-Related Knowledge and Promotion Attitude

Amongst the MSM surveyed population, 64.0% and 69.7% had heard of HPV and HPV vaccine, respectively. About 80.5% knew that men and women could be infected with HPV, and 80.3% knew that men and women could be vaccinated with HPV vaccine. About 67.5% knew that HPV infection could lead to malignant tumors such as condyloma acuminatum, penile cancer, and anal cancer; 73.2% knew that HPV was mainly transmitted through sexual contact; 55.9% knew that the best time to receive the HPV vaccine was before the first sex; 58.4% knew that HPV vaccines could not prevent all types of HPV infections; and only 44.3% knew that most HPV infections had no visible symptoms (Table 4). The HPV-related knowledge level of those who correctly answered more than 9 of the 12 questions was considered to be high, and the rest were medium or low level (Table 5). In addition, 89.9% of the MSM surveyed population supported the promotion of HPV vaccine in China (Table 5).

### 3.5. HPV Vaccination Willingness

Amongst the MSM survey population, 86.21% (350/406) were willing to receive HPV vaccine. Amongst those who were unwilling to vaccinate, 41.1% (23/56) were worried about side effects, 32.1% (18/56) thought that they did not need it, 14.3% (8/56) were worried about economic reasons, 5.4% (3/56) worried about poor results, 3.6% (2/56) worried about information leakage, and 3.6% (2/56) were worried because of other reasons.

### 3.6. Multivariate Analysis of Factors Associated with HPV Vaccination Willingness

In multivariate analysis, HPV vaccination willingness was used as the dependent variable (0 = unwilling, 1 = willing), and variables with *p* < 0.2 in the single-factor analysis were included in the model, including nationality, male sexual partners in the last 6 months, voluntary STD testing, voluntary STD counselling, voluntary HIV counselling, HPV-related knowledge level, HPV infection risk assessment, and HPV infection threat assessment. According to the research results, the education level, marital status and income of the MSM population also had an impact on the population’s willingness to vaccinate with the HPV vaccine. Therefore, education level, marital status, and income were used as adjustment factors.

The results of the study showed that people with high levels of HPV-related knowledge in the MSM population were more willing to receive HPV vaccines than those with low or medium levels (OR = 2.167, 95% CI = 1.049–4.474). People whose HPV infection risk assessment were high were more willing to receive HPV vaccines than those with low or medium levels (OR = 5.905, 95% CI = 1.312–26.580). Moreover, those who supported the promotion of HPV vaccine in the country in the future were more willing to receive the HPV vaccine than those who were neutral or opposed (OR = 6.784, 95% CI = 3.164–14.546) (Table 6).

## 4. Discussion

The MSM population is a high-risk group of HPV infection, and HPV vaccination is undoubtedly a highly cost-effective preventive measure for this population. This study found that 86.21% of the MSM surveyed population were willing to receive HPV vaccine and had a high willingness to vaccinate. According to the theory of KAP, knowledge and beliefs directly or indirectly affect behavior changes. In the same way, learning the relevant knowledge of HPV infection and vaccines can promote the generation of people’s health beliefs, including the risk perception of HPV infection and a positive attitude towards vaccine promotion, which will ultimately affect the willingness to vaccinate against HPV.

In this study, the MSM population with high levels of HPV-related knowledge were more willing to receive HPV vaccine (OR = 2.167, 95% CI = 1.049–4.474), which was consistent with the conclusions of other studies [35,36,37]. The possible reason is that the MSM population has a clearer understanding of the possible adverse consequences of HPV infection and the benefits of HPV vaccination by understanding HPV infection and vaccine-related knowledge, which makes the MSM population more willing to take protective measures because of fear of infection. This result emphasizes the importance of awareness and education in the promotion of HPV vaccine amongst the MSM population. However, only 44.3% of the MSM population had a high level of HPV-related knowledge, and the accuracy of individual questions was less than 50%. This finding showed that the MSM population still had a great lack of understanding of HPV infection and vaccines. This result also suggests that we should develop educational interventions to raise awareness of HPV infection and related diseases in order to improve the acceptability of HPV vaccines.

High levels of HPV infection risk perception is significantly related to HPV vaccination willingness (OR = 5.905, 95% CI = 1.312–26.580). The possible reason is that the MSM population has a better understanding of HPV infection and are more afraid of the undesirable results of HPV infection, so they are more willing to take preventive measures (such as HPV vaccination) to reduce their risk of infection. This reminds us that, in the process of education, we can appropriately emphasize the high incidence of HPV-related diseases and explain in detail the harm caused by HPV-related diseases so as to improve their perception of the risk of HPV infection.

The results of this study also show that the sexual behavior characteristics and preventive behaviors of the MSM population are related to HPV vaccination willingness. People with multiple male sexual partners and MSM who have underwent STD and AIDS consultations are more willing to receive HPV vaccine. The possible reason is that, compared with other people, this part of the MSM population are more likely to be infected with HPV, pay more attention to their own health, and are more worried about the possible negative effects of HPV infection. Therefore, this part of the MSM population have higher risk perception and are more inclined to improve their protection through HPV vaccination. These findings suggest that we can combine HPV vaccine education with STD education in order to increase the willingness of HPV vaccination amongst MSM populations to effectively educate the target population.

Approval of the promotion of HPV vaccine in China is also highly related to the willingness of HPV vaccination (OR = 6.784, 95% CI = 3.164–14.546). At present, the promotion of HPV vaccine amongst the MSM population is a very new phenomenon. People mostly hold the view that MSM are generally unwilling to receive HPV vaccine [38]. In order to correct the general perception of the MSM population, it is necessary to establish a new concept of the acceptability of the HPV vaccine. Health science popularization and media promotion are very effective ways to increase the awareness of HPV infection and vaccines amongst the MSM population, so as to form a positive attitude towards HPV vaccination.

In addition, this study also found that the current MSM population was relatively young (42.6% were 18–30 years old, 34.7% were 30–40 years old) and, at the same time, highly educated (69.5% had college degree or above). These two groups of people have stronger learning ability and are more receptive to new things. Therefore, the promotion of HPV vaccination between these two groups of people should be able to achieve better results. Relevant studies also show that vaccinating with the HPV vaccine before the first sexual intercourse can bring greater benefits, so the promotion of HPV vaccine should be targeted at younger MSM populations [38]. At the same time, we should also pay attention to the differences in the educational level of this group. When adopting educational interventions, we must use simple and clear methods to ensure that regardless of the level of education, it is easy to understand the benefits and importance of HPV vaccination. This is very helpful and vital to the popularization of the HPV vaccine.

China has a large population base. The MSM population is large and has a high HPV infection rate, but there is no corresponding HPV vaccination program. In improving HPV vaccination amongst MSM populations and even the entire country, China faces a very big challenge, and further efforts are needed.

This study recognizes the limitations that should be addressed in follow-up studies. Firstly, this is a cross-sectional study. Although this study shows an association between the independent variables and the dependent variable, it cannot confirm the causality. Secondly, this study only investigated the HPV vaccination willingness of the HIV-negative MSM population, but did not investigate the vaccination willingness of the HIV-positive MSM population. According to existing studies, the HIV-positive MSM population has a higher HPV infection rate, and their willingness and demand for HPV vaccination may be more urgent [20]. Finally, because of the concealment of the MSM population, this study adopted a non-probability sampling method, so the results may be biased, and because it was an electronic questionnaire, some older MSM people could not be surveyed.

## 5. Conclusions

The MSM population has a high willingness for HPV vaccination. Health education for the MSM population should be strengthened, and their awareness of HPV infection and vaccines and the risk of HPV infection should be increased, which will help improve the HPV vaccination willingness amongst the MSM population.

## Figures and Tables

**Table 1 vaccines-09-01069-t001:** Relationship between demographic characteristics and HPV vaccination willingness.

Characteristics	Total (*n* = 406)		HPV Vaccination Willingness		Chi-Square	*p*-Value
	*N*	%	Yes *n* (%)	No *n* (%)		
**Age**					3.0606	0.2165
<30	173	42.6	155 (89.6)	18 (10.4)		
30~	141	34.7	119 (84.4)	22 (15.6)		
40~	92	22.7	76 (82.6)	16 (17.4)		
**Area**					0.6630	0.4155
Urban	273	67.2	238 (87.2)	35 (12.8)		
Rural	133	32.8	112 (84.2)	21 (15.8)		
**Nationality**					-	0.1138 ^a^
Han nationality	397	97.8	344 (86.7)	53 (13.3)		
Other nationalities	9	2.2	6 (66.7)	3 (33.3)		
**Educational level**					0.1189	0.7302
Senior high or below	124	30.5	108 (87.1)	16 (12.9)		
College or above	282	69.5	242 (85.8)	40 (14.2)		
**Employment status**					-	0.8176 ^a^
Employed	303	74.6	259 (85.5)	44 (14.5)		
Retired	7	1.7	6 (85.7)	1 (14.3)		
Students at school	60	14.8	54 (90.0)	6 (10.0)		
Unemployed	36	8.9	31 (86.1)	5 (13.9)		
**Marital status**					1.2919	0.5242
Unmarried	303	74.6	261 (86.1)	42 (13.9)		
Married	67	16.5	56 (83.6)	11 (16.4)		
Divorced or widowed	36	8.9	33 (91.7)	3 (8.3)		
**Personal monthly income (RMB)**					-	0.3381 ^a^
<1000	28	6.9	24 (85.7)	4 (14.3)		
1000~	221	54.4	196 (88.7)	25 (11.3)		
5000~	126	31.0	103 (81.7)	23 (18.3)		
10,000~	31	7.6	27 (87.1)	4 (12.9)		

HPV: human papillomavirus. ^a^ Fisher’s exact test method.

**Table 2 vaccines-09-01069-t002:** Relationship between sexual behavior characteristics, preventive behaviors, and HPV vaccination willingness.

Characteristics	Total (*n* = 406)		HPV Vaccination Willingness		Chi-Square	*p*-Value
	*N*	%	Yes *n* (%)	No *n* (%)		
**Male sexual partners in the past 6 months (including temporary and long-term)**					6.0363	0.0489
0	120	29.6	97 (80.8)	23 (19.2)		
1	174	42.9	150 (86.2)	24 (13.8)		
>1	112	27.6	103 (92.0)	9 (8.0)		
**Whether using condoms during sex** ^b^					1.0408	0.3076
Yes	253	78.1	222 (87.8)	31 (12.2)		
No	71	21.9	59 (83.1)	12 (16.9)		
**Voluntary STD testing**					2.6775	0.1018
Yes	304	74.9	267 (87.8)	37 (12.2)		
No	102	25.1	83 (81.4)	19 (18.6)		
**Voluntary STD counselling**					5.7090	0.0169
Yes	254	62.6	227 (89.4)	27 (10.6)		
No	152	37.4	123 (80.9)	29 (19.1)		
**Voluntary HIV testing**					0.9023	0.3422
Yes	350	86.2	304 (86.9)	46 (13.1)		
No	56	13.8	46 (82.1)	10 (17.9)		
**Voluntary HIV counselling**					9.4586	0.0021
Yes	306	75.4	273 (89.2)	33 (10.8)		
No	100	24.6	77 (77.0)	23 (23.0)		
**Diagnosed with STDs in the past 6 months**					1.1830	0.2767
Yes	55	13.6	50 (90.9)	5 (9.1)		
No	351	86.5	300 (85.5)	51 (14.5)		

HPV: human papillomavirus; STD: sexually transmitted disease; HIV: human immunodeficiency virus. ^b^ Indicating loss of data.

**Table 3 vaccines-09-01069-t003:** Relationship between risk perception and HPV vaccination willingness.

Characteristics	Total (*n* = 406)		HPV Vaccination Willingness		Chi-Square	*p*-Value
	*N*	%	Yes *n* (%)	No *n* (%)		
**HPV infection risk assessment**					14.9897	0.0006
High level	80	19.7	78 (97.5)	2 (2.5)		
Medium level	135	33.3	119 (88.1)	16 (11.9)		
Low level	191	47.0	153 (80.1)	38 (19.9)		
**HPV infection threat assessment**					11.4734	0.0032
High level	173	42.6	160 (92.5)	13 (7.5)		
Medium level	131	32.3	110 (84.0)	21 (16.0)		
Low level	102	25.1	80 (78.4)	22 (21.6)		
**STD infection risk assessment** ^c^					18.7763	<0.0001
High level	47	13.4	47 (100.0)	0 (0.0)		
Medium level	101	28.8	93 (92.1)	8 (7.9)		
Low level	203	57.8	160 (78.8)	43 (21.2)		

HPV: human papillomavirus; STD: sexually transmitted disease. ^c^ In the past six months, people who had not been diagnosed with STDs answered this question.

**Table 4 vaccines-09-01069-t004:** HPV infection and vaccine-related knowledge responses in MSM survey population.

Characteristics	Total (*n* = 406)	
	Correct *n*	%
Heard of HPV	260	64.0
Heard of HPV-related diseases	262	64.5
Using condoms can reduce HPV infection	306	75.4
Men and women can be infected with HPV	327	80.5
HPV infection can lead to condyloma acuminata and malignant tumors such as cervical cancer, penile cancer and skin cancer	274	67.5
HPV is mainly transmitted through sexual contact	297	73.2
Most HPV infections have no visible symptoms	180	44.3
Heard of HPV preventive vaccine	283	69.7
HPV vaccine can effectively prevent cervical cancer	251	61.8
The best time to receive the HPV vaccine is before the first sex	227	55.9
Men and women can be vaccinated with HPV vaccine	326	80.3
HPV vaccine cannot prevent all types of HPV infection	169	41.6

**Table 5 vaccines-09-01069-t005:** Relationship between HPV-related knowledge, promotion attitude, and HPV vaccination willingness.

Characteristics	Total (*n* = 406)		HPV Vaccination Willingness		Chi-Square	*p*-Value
	*N*	%	Yes *n* (%)	No *n* (%)		
**HPV-related knowledge level** ^d^					13.8112	0.0002
High level	180	44.3	168 (93.3)	12 (6.7)		
Medium or low level	226	55.7	182 (80.5)	44 (19.5)		
**Attitudes towards the promotion of HPV vaccine in China in the future**					46.9500	<0.0001
Approve	365	89.9	329 (90.1)	36 (9.9)		
Neutral or opposed	41	10.1	21 (51.2)	20 (48.8)		

HPV: human papillomavirus; STD: sexually transmitted disease. ^d^ Answering more than nine questions correctly indicates a high level of knowledge, and the rest are medium or low levels.

**Table 6 vaccines-09-01069-t006:** Multivariate logistic regression analysis of HPV vaccination willingness in MSM.

Characteristics	OR	95% CI	*p*-Value
**HPV-related knowledge level**			
High level	2.167	1.049–4.474	0.0366
Medium or low level (reference)			
**HPV infection risk assessment**			
High level	5.905	1.312–26.580	0.0461
Medium level	1.698	0.859–3.357	0.4479
Low level (reference)			
**Attitudes towards the promotion of HPV vaccine in China in the future**			
Approve	6.784	3.164–14.546	<0.0001
Neutral or opposed (reference)			

Adjust education level, marital status and income.

## Data Availability

The datasets involved in the current study are not publicly available due to privacy but are available from the author Wei He on reasonable request.

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
