# Peer review of "Analysis of HPV Vaccination Willingness amongst HIV-Negative Men Who Have Sex with Men in China"

_vaccines, 2021, doi:10.3390/vaccines9101069_

Round 1

Reviewer 1 Report

I consider that this paper has new findings and is worth publishing. However, I would like to suggest 4 minor concerns to be revised.

  1. The definition of HIV-negative is ambiguous. Related to it, did the authors asked the study participants about the frequency of voluntary STD testing and voluntary HIV testing?
  2. There are many repetitions of the descriptions about the results in the discussion. Instead, I recommend adding information on China's national immunization program policy or the status on the HPV vaccine and discuss the results.
  3. Did the questionnaire include questions about oropharyngeal cancer? In many countries, there are concerns about an increase in young male HPV-related oropharyngeal cancers. How about in China? If not, please explain the reason why the authors didn’t include the topic.
  4. As a supplementary material, please disclose the questions of all the questionnaires that were actually used.

Reviewer 2 Report

An interesting manuscript investigating the perceptions for HPV vaccination in a sample of MSM community in China.

There are a few comments

  1. “In this study, frequency and percentage were used to describe the data, and the chi-square test or Fisher’s exact test was used for count data.” perhaps the authors mean to compare groups for categorical data?
  2. “Multivariate analysis adopted 108 stepwise logistic regression analysis”, please provide more details: for example if you have used a forward selection process or backward elimination.
  3. “2.3. Statistical analysis”, please transfer the sample size calculation in this section, and additionally would be nice to provide more details for the assumptions, tests, and tools used for sample size calculation, thus the interesting reader can reproduce the calculations.
  4. The authors have already declared that have used a non-probability sampling method, this is a disadvantage of the research, and possible issues in the results should be added in the discussion (for example in electronic-only questionnaires, higher age population can be excluded if they do not have access to computers).

Reviewer 3 Report

The manuscript number 1388720 entitled “Analysis of  human papillomavirus (HPV) vaccination willingness amongst HIV-negative men who have sex with men in China” investigates the HPV vaccination willingness of Men who have sex with men (MSM) and related factors that affect the vaccination willingness, and provide a reference for the future development of HPV vaccination strategies for male populations. This study was based on a questionnaire, including sociodemographic characteristics, sexual behaviour characteristics, HPV infection and vaccine-related knowledge, risk perception, and HPV vaccination willingness and promotion attitudes. This is an interesting study but needs some improvements before to be considered for publication.

The main question is related to following information. Why the authors mention that this study explored the willingness of human immunodeficiency virus (HIV)-negative MSM for HPV vaccination? It is not clear what is the relation of HIV-negative MSM with HPV infection and HPV vaccination, and should be clarified in the abstract, introduction and discussion.

In line 46 it is referred that the prevalence of HPV amongst women aged 18-24 years dropped from 22.7% to 1.1% in 2005-2015. But what demographic region is this situation? It is relevant also mentioned that in developed countries this is the normal scenario and in fact the HPV infection incidence significantly decreased, but the same did not occur in the low income countries (LIC). The LIC have poor health systems, who fail to access of the HPV vaccine to the population, and also the anti-vaccination movement decrease the vaccination rates, being the HPV infection  more disseminated and consequently the appearance of HPV-derived diseases and malignances. This situation should also be reflected in the introduction section.    
